

# A machine learning assistant for detecting fraudulent activities in synchronous online programming exams

Francisco Ortin[1,2], Alonso Gago[1], Jose Quiroga[1] and Miguel Garcia[1]

[1] Computer Science, Universidad de Oviedo, Oviedo, Asturias, Spain
[2] Computer Science, Munster Technological University, Cork, Ireland

## ABSTRACT

The rapid expansion of online learning has made education more accessible but has also introduced significant challenges in maintaining academic integrity, particularly during online exams. For certain types of exams, students are prohibited from connecting to the Internet to prevent them from accessing unauthorized resources, utilizing generative artificial intelligence tools, or engaging in other forms of cheating. However, in online exams, students must remain connected to the Internet. Most existing online proctoring systems rely on various devices to monitor students' actions and environments during the exam, focusing on tracking physical behavior, such as facial expressions, eye movements, and the presence of unauthorized materials, rather than analyzing the students' work within their computers. This often requires human review to determine whether students are engaging in unauthorized actions. This article presents the development and evaluation of a machine-learning-based assistant designed to assist instructors in detecting fraudulent activities in real-time during online programming exams. Our system leverages a convolutional neural network (CNN) followed by a recurrent neural network (RNN) and a dense layer to analyze sequences of screenshot frames captured from students' screens during exams. The system achieves an accuracy of 95.18% and an $F_2$-score of 94.2%, prioritizing recall to emphasize detecting cheating instances, while minimizing false positives. Notably, data augmentation and class-weight adjustments during training significantly enhanced the model's performance, while transfer learning and alternative loss functions did not provide additional improvements. In post-deployment feedback, instructors expressed high satisfaction with the system's ability to assist in the rapid detection of cheating, reinforcing the potential of machine learning to support real-time monitoring in large-scale online exams.

# INTRODUCTION

The rapid growth of online learning has fundamentally transformed traditional educational models, enabling a more flexible and inclusive approach to education (*Ally, 2004*). Distance education, powered by digital tools, provides significant advantages in terms of accessibility. It removes geographic barriers, allowing students to participate in

Corresponding author
Francisco Ortin, ortin@uniovi.es

courses from anywhere in the world. This opens doors to learners from diverse backgrounds, granting them access to high-quality education without the necessity of being physically present on a campus.

Delivering effective synchronous online programming labs poses significant challenges, particularly due to the need for real-time interaction and code monitoring between instructors and students—similar to traditional face-to-face labs (*Fita et al., 2016*). In response to these challenges, we developed a remote synchronous infrastructure that enables students to participate in remote labs from their own devices while instructors monitor their coding activities in real time (*Garcia, Quiroga & Ortin, 2021*). Built on a modified computer monitoring system, virtual private network (VPN) access, and automation scripts for easy deployment and management, the infrastructure also integrates web conferencing tools to facilitate communication and instructional support. This system has been successfully used in programming courses and flipped learning settings, demonstrating high student satisfaction and efficient resource use (*Ortin, Quiroga & Garcia, 2023*).

A significant challenge in distance learning is ensuring academic integrity during online exams (*Atoum et al., 2017*). In some cases, students are prohibited from connecting to the Internet during exams to prevent access to unauthorized resources, generative artificial intelligence tools, or other means of cheating. However, the necessity of an Internet connection for conducting online exams creates a paradox, as students are simultaneously required to be online while needing to be restricted from accessing certain online materials. To address this issue, various online proctoring systems have been developed aimed at monitoring student behavior and upholding exam integrity (*Noorbehbahani, Mohammadi & Aminazadeh, 2022*). More advanced systems integrate machine learning (ML) algorithms to analyze student behavior patterns and flag potential instances of cheating for further review by human proctors (*Gopane et al., 2024*). However, these systems typically rely on real-time video surveillance to detect suspicious actions (next section), and do not commonly analyze the students' activities within their computers.

In addition to standard measures such as webcam activation and real-time instructor supervision, our infrastructure supports continuous monitoring of students' work throughout online exams. However, for courses with large enrollments—with up to 200 students—it becomes increasingly difficult for instructors to detect fraudulent activities during the exam. To address this challenge, we have developed a machine-learning system to assist instructors in identifying potential instances of cheating in synchronous online programming exams. This ML assistant analyzes screenshot frames from students' screens, captured by our infrastructure, and alerts the instructor to potential fraudulent activities. The potentially fraudulent screenshot sequence detected by the assistant is shown to the instructor (Fig. 1), who can either dismiss the alert if it is a false positive or take appropriate action, such as sending a popup message to the student *via* the system.

The primary contribution of this work is the development and implementation of an ML-based proctoring system that assists instructors in effectively monitoring large-scale online exams, thereby safeguarding academic integrity and reducing the risk of cheating. This system requires no additional hardware beyond a computer connected to the Internet

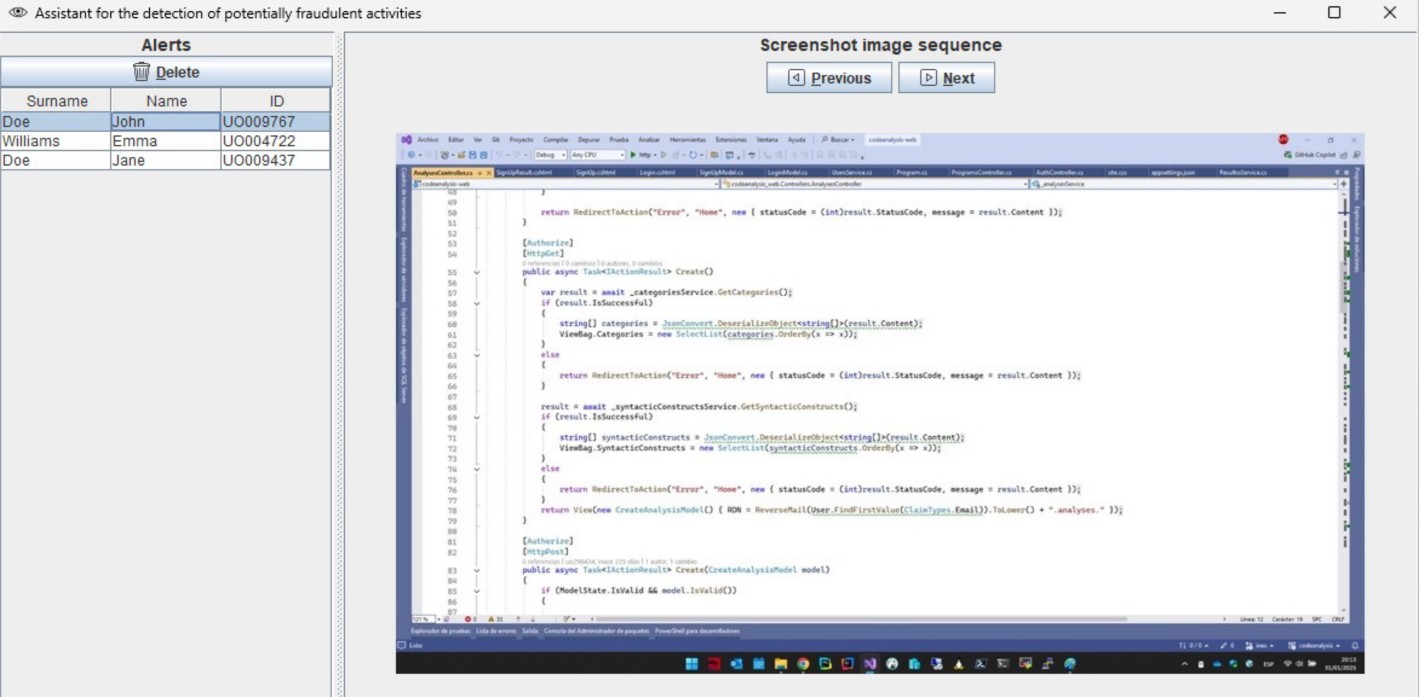

**Figure 1** ML assistant showing alerts of potentially fraudulent activities.     

and is focused on analyzing students' activities within their computers. It can be complemented with other online proctoring systems that utilize additional hardware to monitor students' actions and environments during the exam.

## RELATED WORK

*Migut et al. (2018)* propose a system with objectives similar to those presented in this article. They conducted preliminary controlled experiments in which screen video recordings of four exams were collected from two students. The participants deliberately introduced predefined instances of unauthorized behavior according to a pre-established protocol. After the exams, the videos were analyzed using visual similarity between successive frames—a kind of shot segmentation technique (*Yuan et al., 2007*)—to detect changes in screen content, such as those caused by switching applications. By identifying these changes, the proctor no longer needs to review the entire video. A key difference between Migut's work and ours is that their approach does not support real-time intervention to warn students about potential consequences, thereby missing the opportunity to prevent ongoing fraudulent behavior. Moreover, they did not report any measurements of the algorithm's accuracy in detecting fraud. Their work remains at the proposal stage, with no implementation or reported results (*Migut et al., 2018*).

*Smirani & Boulahia (2022)* propose a system based on a convolutional neural network (CNN) that does not require the use of a webcam. Their model is trained using students' personal and geographical data, along with screenshots of their activity during the exam. This approach is limited to online tests conducted within the Blackboard Learning

Management System (LMS), and it is capable of detecting only one type of fraudulent behavior: Internet browsing. These constraints define a significantly more restrictive setting compared to the scenario considered in our study, which focuses on online programming exams. Under these conditions, *Smirani & Boulahia (2022)* report an accuracy of 98.5%, a recall of 99.8%, and a precision of just 1.8%. The system does not support any form of real-time intervention.

Many existing systems for detecting fraudulent activity in online exams are based on monitoring students' actions and environments during the exams, commonly referred to as online proctoring systems (*Noorbehbahani, Mohammadi & Aminazadeh, 2022*). These systems utilize various proctoring technologies (such as identity authentication, webcam monitoring, and keystroke recognition) to track and analyze student behavior in real time to detect suspicious or abnormal activities that could indicate cheating or dishonesty. These approaches typically require additional hardware to the two screen-monitoring systems discussed above and the one presented in this article.

One of the most widely used methods for online exam proctoring is camera monitoring. *Luan, Ha & Hung (2022)* propose a system where students are recorded with two different cameras. The first camera captures images of a student's face to detect anomalies in their movements, while the second camera records their body and the surrounding environment. By using a trained pose recognition model, the system can efficiently classify student actions as suspicious or not. ProctorExam further augments spatial controls by utilizing innovative 360-degree monitoring (*Turnitin, 2025*). This system includes webcam monitoring, screen-sharing, and a smartphone camera positioned to view everything around the student. *Atoum et al. (2017)* propose a system using one webcam, one wearable camera, and a microphone to monitor the visual and acoustic environment of the testing location.

Machine and deep learning techniques have also been leveraged to assist instructors in detecting cheating behavior (*Nigam et al., 2021*). *Gopane et al. (2024)* trained a deep-learning model that detects patterns in students' head and eye movements to assist in cheat detection. They trained CNN models with a dataset of 1,000 online examinations, achieving an $F_1$-score of 0.94. Examus gathers data from students' behavioral characteristics during online lectures and uses this data to enhance proctoring services during online exams (*Examus, 2025*). ProctorNet uses the pre-trained Inception CNN model to detect suspicious behaviors based on students' eye gaze and mouth movements (*Tejaswi, Venkatramaphanikumar & Kishore, 2023*).

Several other online proctoring tools implement student authentication systems. For instance, ProctorU requires students to present their ID cards to the webcam for authentication (*Milone et al., 2017*). Additionally, students must maintain an uninterrupted audio-visual connection with the proctor throughout the session. *Joshy et al. (2018)* implemented a three-fold student authentication scheme based on face recognition, one-time password verification, and fingerprint authentication.

TeSLA aims to develop techniques for verifying students *via* biometrics during online tests (*Draaijer, Jefferies & Somers, 2018*). This includes facial recognition, voice recognition, and keystroke and fingerprint analysis to ensure that no impersonation occurs

and that the answers are provided by the actual test-taker. Other biometric technologies, such as fingerprint scanning, iris scanning, retina scanning, hand scanning, and facial scanning, have been utilized for user authentication in online proctoring systems (*Mahadi et al., 2018*).

Proctorio (*Bergmans et al., 2021*) and ExamSoft (*Karibyan & Sabnis, 2021*) are two widely adopted online proctoring solutions that focus on maintaining exam integrity through continuous monitoring and post-exam analysis. Proctorio employs a combination of webcam, microphone, and screen activity monitoring, using rule-based systems and some AI components to flag behaviors such as gaze aversion, multiple faces, or tab switching. ExamSoft, through its ExamMonitor feature, similarly uses webcam recordings and automated behavior flagging for instructor review after the exam concludes. However, neither system incorporates deep learning models that analyze the temporal sequence of screen content, nor do they support real-time interventions during an exam.

## REMOTE SYNCHRONOUS INFRASTRUCTURE

As previously described, we developed an infrastructure during the COVID-19 lockdown to support remote synchronous programming labs (*Garcia, Quiroga & Ortin, 2021*). Since then, it has been adapted to enable remote access to face-to-face labs and support various flipped learning scenarios (*Ortin et al., 2024*). The system allows students to participate from their own Internet-connected computers while instructors monitor their activities in real time (Fig. 2). Student monitoring is achieved through a customized fork of the Veyon project (*Junghans, 2025*), which enables both live screen viewing and video/frame-sequence recording of all participants (*Garcia, Quiroga & Ortin, 2025*). Veyon comprises two components: the Veyon Service, running on student machines to transmit screen data, and the Veyon Master, a graphical interface used by instructors to monitor and interact with students' screens.

Veyon's client monitoring system offers strong protection against fake image attacks through the use of cryptographic authentication (based on key pairs), encrypted communication, and strict access controls that permit only authorized clients to connect (*Junghans, 2025*). Its architecture employs network isolation, process separation, and internal proxies to validate and securely manage data. In addition, multi-layered authorization mechanisms and regularly applied security updates help safeguard against tampering and unauthorized data injection. While these features make it difficult for students to transmit fake images to the server in an attempt to evade detection during monitoring, any client-side solution may still be susceptible to manipulation by technically skilled users.

Both instructors and students connect to the system over the Internet using a VPN, which places all participants within the same private network. This approach simplifies student access by removing the need for router configuration (*e.g.*, port forwarding) and ensures that the Veyon Master can reliably access student IP addresses, even when dynamic IPs are assigned by residential ISPs.

A set of scripts was developed to streamline key tasks for both students and instructors, including installing, starting, stopping, and uninstalling the remote lab environment, as

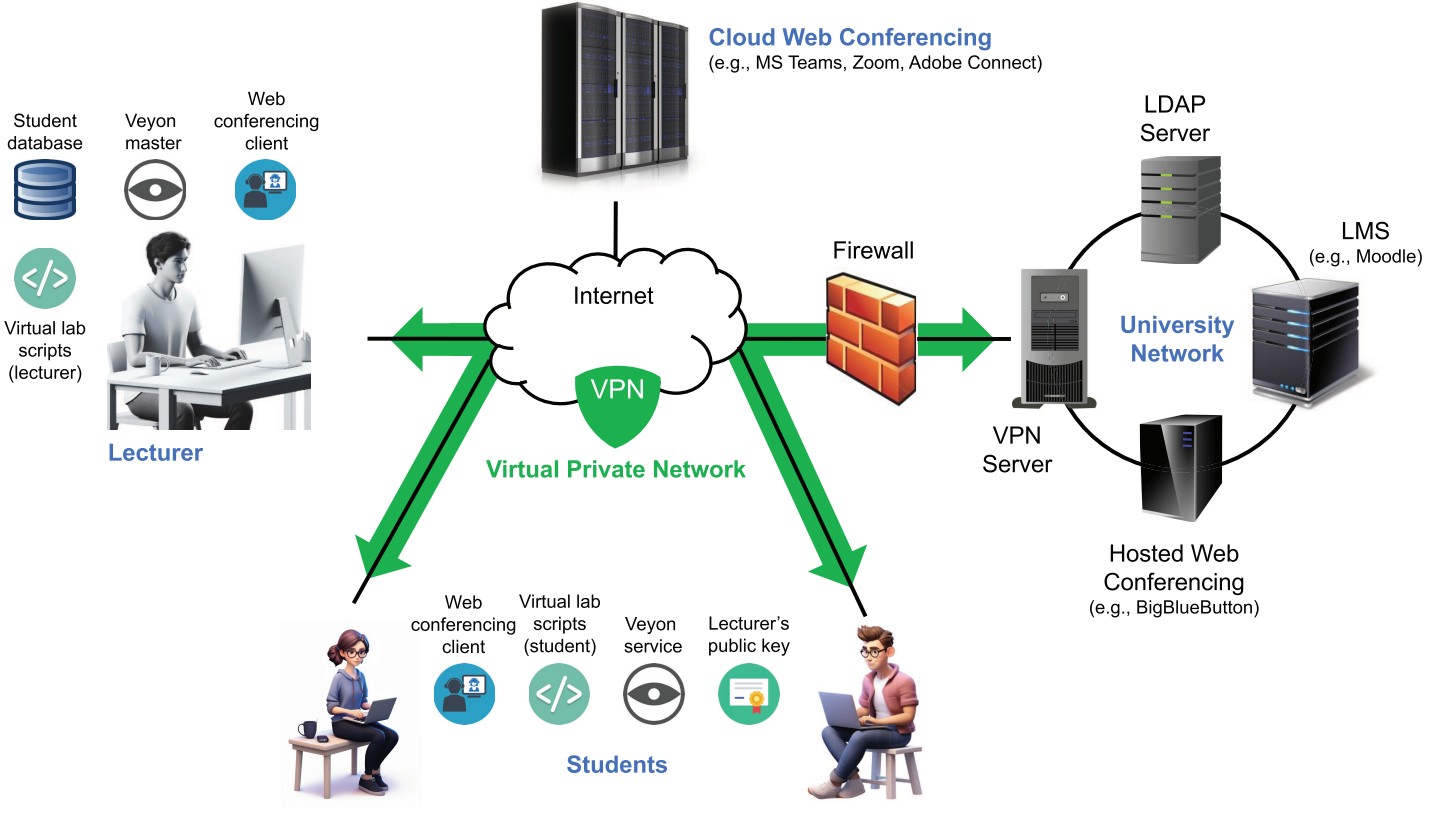

**Figure 2  Architecture of the remote synchronous infrastructure.**

well as retrieving student participation data (instructor only) (*Quiroga, Garcia & Ortin, 2022*). Implemented for both Windows and Linux, the scripts automate operations such as installing the instructor's public key, configuring firewall rules for Veyon, obtaining student information from the VPN server, and setting up lab sessions.

Communication between students and instructors is primarily handled through a web conferencing platform. We used both Microsoft Teams (cloud-hosted) and a university-hosted instance of BigBlueButton. A single platform is used per session to deliver lectures or labs, enabling real-time interaction *via* audio, video, chat, and screen sharing. It also supports private communication, resource sharing, student submissions, and the use of a digital whiteboard (*Bower, 2011*).

## ARCHITECTURE OF THE ASSISTANT

Figure 3 illustrates the architecture of the assistant designed to detect potentially fraudulent activities during online exams—the main contribution of this article. Multiple students may participate in an exam, connected to our remote synchronous infrastructure *via* the Internet. The infrastructure is configured to capture one screenshot per second of each student's activity, storing a sequence of three consecutive frames as input for the machine learning assistant. The assistant system analyzes these input sequences to detect

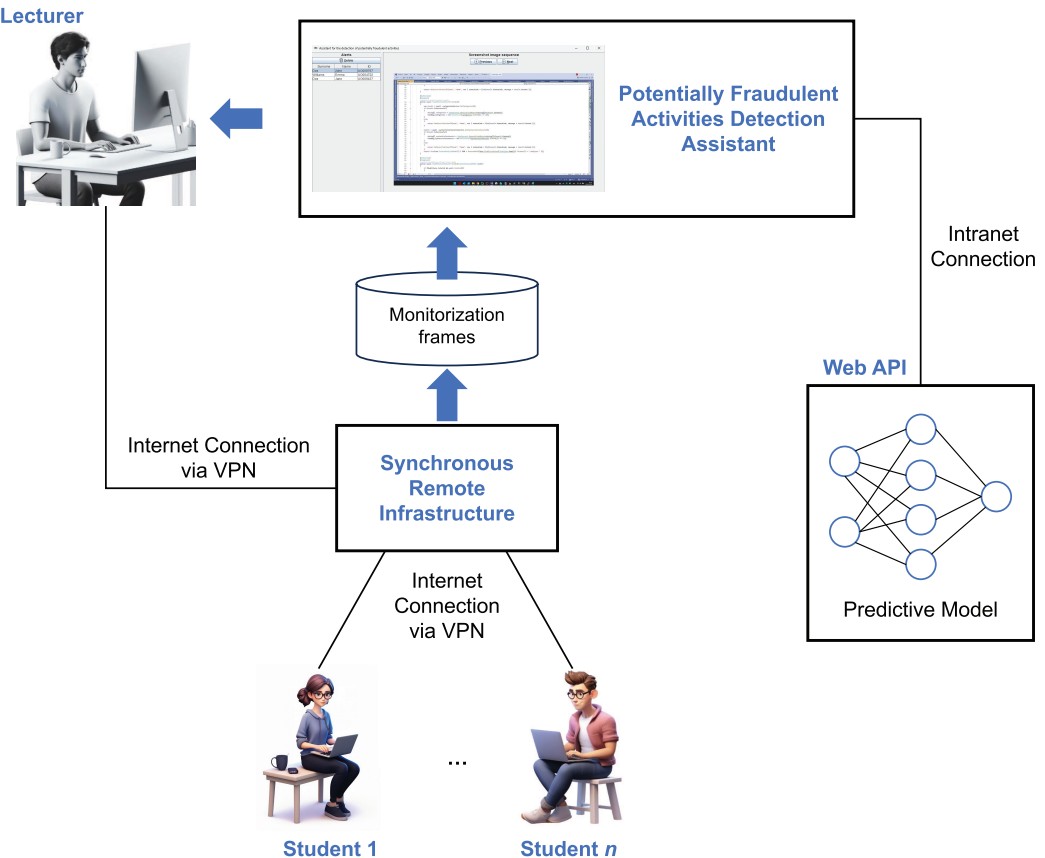

**Figure 3** **Architecture of the assistant designed to detect potentially fraudulent activities during online exams.**

potentially fraudulent behavior, generating an alert only when suspicious activity is first detected after a previously correct screenshot. This approach helps minimize unnecessary alerts, making it easier for the instructor to monitor the students effectively.

Once a sample of three consecutive frames is collected for a given student, the assistant passes the sample to the ML model, which has been trained to identify potentially fraudulent activities (as described in the next section). This model is deployed as a web API implemented in Flask, loading the artificial neural network (ANN) into memory at startup to optimize inference performance. The model functions as a binary classifier, returning "1" if fraudulent activity is detected and "0" otherwise.

If potentially fraudulent activity is detected, the assistant triggers an alert, as shown in Fig. 1, accompanied by a beep sound. The instructor should then revise the screenshots that triggered the alert and decide on the appropriate action. In the case of a false positive, the alert can be dismissed. Otherwise, the student is warned through a popup message sent *via* the remote synchronous infrastructure or through an oral warning. In cases of repeated fraudulent behavior, further actions may be taken, such as suspending the student's ability to continue the exam or escalating the issue to academic authorities for further

investigation. Additionally, the suspicious activity videos recorded by the infrastructure are saved in their original full-size format, providing a detailed record for both the student and the institution to reference, if necessary.

# METHODOLOGY

In this section, we outline the training and evaluation process of the ML model designed to assist in detecting fraudulent activities. We provide an overview of the course context, dataset construction, architecture of the ANNs employed, evaluation metrics, hyperparameter optimization, model training, and the assessment of instruction satisfaction.

## Context

The system was utilized for online exams in the "Programming Technology and Paradigms" course of the Software Engineering degree at the University of Oviedo, Spain (*Ortin, Redondo & Quiroga, 2017*). In this course, students learn object-oriented and functional programming paradigms, concurrent and parallel programming, and basic meta-programming concepts in dynamic languages (*Ortin, Redondo & Quiroga, 2016*). The course consists of 58 class hours (30 h for programming labs and 28 h for lectures) during the second semester, earning six ECTS credits. A total of 181 students were enrolled in the course, of whom 146 were male (80.7%) and 35 female (19.3%). Among them, 69 students (38.1%) were retaking the course. The average student age was 22.6 years (mode: 21; standard deviation: 2.9). The exams involve programming tasks in C#, incorporating both object-oriented and functional paradigms, with an emphasis on sequential and concurrent programming approaches.

## Dataset

To create the dataset, we recorded the activity of various students in different labs, including no personal information—data is fully anonymized. Most of the time, students performed permitted activities with a high variance in screenshots, such as writing code, browsing the desktop, navigating files, and editing text. The system was designed to account for variations such as different desktop configurations, light/dark mode settings, system colors, and screen resolutions. By accounting for these variations, the system can better differentiate between legitimate user actions and potential fraudulent behavior, minimizing false positives and improving overall accuracy.

To enrich the dataset, we simulated common fraudulent activities prohibited during exams, such as internet browsing, use of generative artificial intelligence (AI) tools, unauthorized device interactions, and seeking external assistance *via* messaging or video calls. These simulated behaviors were carefully crafted to resemble real-world cheating scenarios, enabling us to train and evaluate the model's ability to detect a variety of violations in a controlled environment.

After collecting the screenshot frames, we need to label the image sequences. For that, we implemented a simple but useful sequence labeling Java application. This application facilitated the rapid labeling process by allowing users to navigate through the image

sequences and categorize them as either cheating or non-cheating behavior. The labeled dataset was then used for training the model.

The final dataset comprised 8,329 sequences, each consisting of three images with three color channels (5,830 sequences for training and 2,499 for testing). To improve model performance, we augmented the training dataset using six different transformations: flipping, rotation, zooming, cropping, and modifications to contrast and brightness. This resulted in an expanded training dataset of 40,810 sequences. Since the training dataset was imbalanced, we applied undersampling to the majority class (non-cheating), achieving a balanced dataset of 36,702 samples.

We explored the use of conditional generative adversarial networks (GANs) to generate synthetic samples for the underrepresented cheating class, implementing both fully connected and convolutional variants (*Ribas, Casaca & Fares, 2025*). However, the generated images lacked visual and semantic fidelity, failing to capture the structural complexity of real exam screenshots. Due to the poor quality and limited utility of these synthetic samples, they were not used to augment the dataset or train the final model.

## Predictive models

Figure 4 illustrates the topology of the artificial neural network (ANN) used to build the model. Each frame in the input sequence is passed through a series of CNN layers. The number of layers, as well as the number of convolutional filters, kernel sizes, and strides, are hyperparameters defined in the "hyperparameter search and model training" subsection below. The number of convolutional filters doubles with each subsequent layer, following conventional design principles (*Li et al., 2021*).

After processing the frames through the CNN layers, the sequence is fed into a recurrent neural network (RNN) to capture the temporal dependencies between frames. Specifically, we employed either a long short-term memory (LSTM) or a gated recurrent unit (GRU) network (another hyperparameter) to see which better captured the long-range dependencies in the input sequence. The number of recurrent units in the RNN is another hyperparameter.

To prevent overfitting, a dropout layer is applied following the RNN layer, randomly setting a fraction of the recurrent units to zero during training. The dropout rate is a hyperparameter fine-tuned to balance model complexity and generalization performance. The output is then passed through a fully connected (dense) layer that aggregates the features learned by the CNN and RNN components. Finally, a sigmoid activation function is used to produce a binary classification output (fraudulent or non-fraudulent).

We also employed transfer learning to potentially enhance model performance by leveraging pre-trained weights from an image classification task (*Yosinski et al., 2014*). Specifically, we tested the performance of replacing the convolutional layers in Fig. 4 with the pre-trained models detailed in the 'hyperparameter search and model training' subsection. The final classification layer from these pre-trained networks was excluded, and the weights were frozen to retain low-level feature representations. The remaining layers (RNN, dropout, and dense) were trained from scratch. This approach helps the

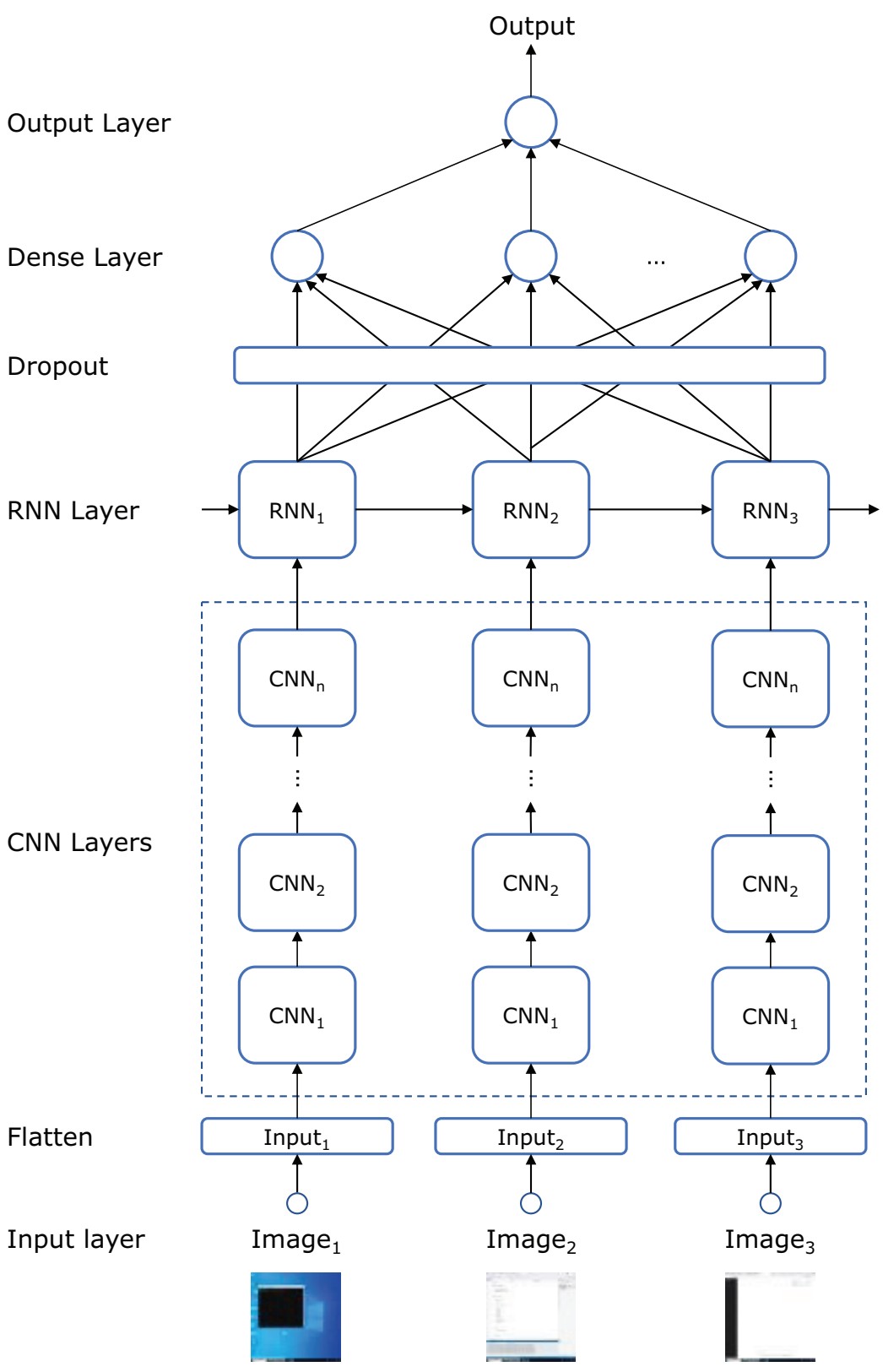

**Figure 4 ANN topology that combines CNN, RNN, dropout, and dense layers.**

model benefit from previously learned low-level features, reducing the need for a large training dataset.

To provide a meaningful baseline for comparison, we included a model that computes traditional optical flow between consecutive screenshots, followed by CNN layers to classify potential cheating behavior (*Tu et al., 2019*). Optical flow is a classical technique used to estimate pixel-wise motion between frames (*Zhai et al., 2021*) and is effective for capturing abrupt visual changes such as tab switches, window resizing, or fast navigation between screenshots—activities that may correlate with unauthorized behavior during online exams. By feeding the resulting motion maps into a CNN, this baseline can leverage spatial features from the motion data without explicitly modeling the temporal sequence (*Tu et al., 2019*)—after the CNN layers, a dense network is added. Although this method does not incorporate long-term dependencies or contextual patterns across multiple screen states, it offers a computationally simpler alternative to our CNN+RNN architecture. We evaluated the *Farnebäck (2003)*, *Lucas & Kanade (1981)*, and *Horn & Schunck (1981)* optical flow algorithms.

## Evaluation metrics

As previously mentioned, the training set was used for model training, while the validation set was used for hyperparameter tuning. Model evaluation was performed on the test set, which was kept separate from both the training and validation sets.

We utilized the following evaluation metrics, where TP (true positives) refers to the number of samples correctly classified as fraudulent, TN (true negatives) is the number of samples correctly classified as non-fraudulent, FP (false positives) refers to the number of samples incorrectly classified as fraudulent, and FN (false negatives) represents the number of samples incorrectly classified as non-fraudulent:

– Accuracy (Eq. (1)). Accuracy measures the overall correctness of the model's predictions, calculated as the proportion of correct predictions to total predictions.

$$\text{Accuracy} = \frac{TP + TN}{TP + TN + FP + FN} = \frac{\text{Number of correct predictions}}{\text{Total number of predictions}}. \tag{1}$$

– Precision (Eq. (2)) measures the proportion of true positives among all positive predictions, measuring the accuracy of fraudulent predictions.

$$\text{Precision} = \frac{TP}{TP + FP}. \tag{2}$$

– Recall (Eq. (3)) calculates the ratio of true positives to all actual positive cases, quantifying the model's ability to identify fraudulent activities.

$$\text{Recall} = \frac{TP}{TP + FN}. \tag{3}$$

– $F_1$-score. The harmonic mean of precision and recall, providing a single value that balances both metrics. $F_1$-score is a special case of the $F_\beta$-score (Eq. (4)) with $\beta = 1$, giving equal weight to both precision and recall. The $F_\beta$-score allows for the adjustment of the balance between precision and recall by modifying $\beta$, where a higher $\beta$ favors

recall and a lower $\beta$ favors precision, providing a flexible metric for various applications.

$$F_{\beta}\text{-score} = (1 + \beta)^2 \cdot \frac{\text{precision} \cdot \text{recall}}{\beta^2 \cdot \text{precision} + \text{recall}}. \qquad (4)$$

– $F_2$-score. Similar to the $F_1$-score but with twice the weight on recall ($\beta = 2$), reflecting the higher cost of false negatives (missing fraudulent activities) compared to false positives (incorrectly flagging non-fraudulent activities as fraudulent). $F_2$-score is particularly relevant in this problem, where detecting fraudulent activities is more critical than minimizing false positives, even if it increases the false positive rate.

## Hyperparameter search and model training

The topologies identified involve several hyperparameters that significantly influence the network's learning and generalization ability. Some hyperparameters pertain to the architecture (*e.g.*, the number of layers, neurons per layer), while others relate to the training process (*e.g.*, batch size, learning rate).

To leverage transfer learning, we experimented with several pre-trained CNN models known for their performance with small images: ResNet50 (*He et al., 2016*), MobileNetV2 (*Sandler et al., 2018*), EfficientNetB0 (*Tan & Le, 2019*), NASNetMobile (*Zoph et al., 2018*), VGG16, and VGG19 (*Simonyan & Zisserman, 2014*). These models were pre-trained on the ImageNet dataset, allowing us to use their learned low-level feature extraction capabilities. The selection of these models was based on their established success in a variety of computer vision tasks, particularly image classification, and their diverse architectural approaches, ranging from deep residual networks to lightweight models designed for mobile devices.

We conducted an exhaustive grid search to identify the best hyperparameter combinations. In each iteration, the model was trained with the training set using a specific hyperparameter set, and performance was assessed using the $F_2$-score on the validation set (Eq. (4)). We explored various optimizers (Adam, RMSprop, Adamax) to determine the best one for training.

Two training approaches were tested to improve the $F_2$-score:

1. Assigning twice the weight to the positive class (fraudulent activities) to encourage the model to minimize false negatives (*Cui et al., 2019*). This adjustment encourages the model to be more sensitive to detecting fraudulent activities, even at the cost of increasing false positives (Eq. (4)).
2. Using an alternative loss function. In addition to the typical binary cross-entropy loss function for binary classification problems, we tried a differentiable surrogate loss function designed to optimize $F_2$-score (*Lee, Yang & Yoo, 2021*).

After completing the hyperparameter search process, we selected the set of hyperparameters that yielded the best $F_2$-score performance. Tables 1, 2 and 3 provide the values used for each hyperparameter in, respectively, CNN+RNN, transfer learning, and optical flow models, along with the values corresponding to the best-performing

**Table 1  Best hyperparameters found for the CNN+RNN neural network.**

| Hyperparameter | Range | Result |
| --- | --- | --- |
| Convolutional layers | [1, 2, 3] | 1 |
| Convolutional filters | [4, 8, 16, 32] | 8 |
| Convolutional kernel size | [3, 4, 5] | 3 |
| Strides | [1, 2] | 1 |
| Convolutional activation function | [ReLU, Leaky ReLU] | ReLU |
| RNN cells | [LSTM, GRU] | LSTM |
| RNN activation function | [Tanh, ReLU] | Tanh |
| RNN units | [16, 32, 64, 128] | 64 |
| Hidden dense layer | [0, 1, 2] | 1 |
| Units in hidden dense layer | [16, 32, 64, 128] | 64 |
| Hidden layer activation function | [ReLU, Leaky ReLU] | ReLU |
| Dropout rate | [0.0, 0.2, 0.5] | 0.5 |
| Learning rate | $[10^{-4}, 10^{-3}, 10^{-2}, 10^{-1}]$ | $10^{-3}$ |
| Batch size | [32, 64, 128] | 64 |
| Optimizer | [Adam, RMSprop, Adamax] | RMSprop |

**Table 2  Best hyperparameters found for the transfer learning approach.**

| Hyperparameter | Range | Result |
| --- | --- | --- |
| Pre-trained model | [EfficientNetB0, MobileNetV2, NASNetMobile, ResNet50, VGG16, VGG19] | EfficientNetB0 |
| RNN cells | [LSTM, GRU] | LSTM |
| RNN activation function | [Tanh, ReLU] | Tanh |
| RNN units | [16, 32, 64, 128] | 32 |
| Hidden dense layer | [0, 1, 2] | 0 |
| Units in hidden dense layer | [16, 32, 64, 128] | 32 |
| Hidden layer activation function | [ReLU, Leaky ReLU] | ReLU |
| Dropout rate | [0.0, 0.2, 0.5] | 0.2 |
| Learning rate | $[10^{-4}, 10^{-3}, 10^{-2}, 10^{-1}]$ | $10^{-3}$ |
| Batch size | [32, 64, 128] | 32 |
| Optimizer | [Adam, RMSprop, Adamax] | Adam |

**Table 3  Best hyperparameters found for the optical flow approach.**

| Hyperparameter | Range | Result |
| --- | --- | --- |
| Optical flow algorithm | [Farneback, Lucas-Kanade, Horn-Schunck] | Farneback |
| CNN layers | [1, 2, 3] | 3 |
| Units in dense layer | [32, 64, 128, 256, 512] | 256 |
| Dropout rate | [0.0, 0.2, 0.5] | 0.5 |
| Learning rate | $[10^{-4}, 10^{-3}, 10^{-2}, 10^{-1}]$ | $10^{-3}$ |
| Batch size | [32, 64, 128] | 32 |
| Optimizer | [Adam, RMSprop, Adamax] | Adam |

configurations. For each iteration of the search, the models were trained for a maximum of 10 epochs, with early stopping applied based on $F_2$-score performance and a patience of two epochs.

Regarding initialization methods, we applied Xavier (Glorot) initialization to the convolutional layers and the final output dense layer (*Glorot & Bengio, 2010*), and He initialization to the recurrent and hidden dense layers (*He et al., 2015*).

Once the optimal set of hyperparameters was determined, we conducted a final fine-tuning training session. In the case of transfer learning, the weights of the pre-trained models were unfrozen to enable fine-tuning on the target task. This adjustment ensured that the pre-trained features were adapted to better align with the specific domain of the problem. The stopping criterion for fine-tuning in both topologies was based on an increase in validation loss over three consecutive epochs. Additionally, the learning rate was dynamically reduced by a factor of 0.2 if the validation loss did not improve in the last epoch. The model exhibiting the best $F_2$-score performance on the validation set was selected for final evaluation with the test set.

To assess the statistical significance of the results, we employed bootstrapping with 10,000 repetitions to calculate 95% confidence intervals for each evaluation metric (*Davison & Hinkley, 1997*). This process involved repeatedly sampling with replacement from the test set to generate multiple resampled datasets. For each resample, the metrics were computed for both models, yielding paired distributions of metric values across all 10,000 bootstrapped samples. Confidence intervals for the average of each metric were then derived from these distributions. In addition to the confidence intervals, we conducted paired *t*-tests on the bootstrapped metric values to assess whether the differences between the models were statistically significant at $\alpha = 0.05$ (*Georges, Buytaert & Eeckhout, 2007*). The paired structure ensures that each comparison reflects performance on the same underlying data sample, thereby accounting for the correlation between resampled estimates and improving the reliability of the significance testing.

Hyperparameter search and model training were performed on a Dell PowerEdge T630 equipped with two Intel Xeon E5-2630 processors (2.4 GHz), 128 GB of RAM, and an NVIDIA Geforce GTX 1050 Ti GPU with 4GB GDDR5, running an updated 64-bit version of Windows 11.

## Image size

As shown in Fig. 3, the synchronous remote infrastructure must be configured to store monitoring frames, which are later used by the assistant to detect potentially fraudulent activities. A key decision involves selecting an appropriate image size: the images must be small enough to allow real-time processing by the predictive model—so that instructors can be alerted promptly—yet large enough to enable accurate predictions.

To determine the optimal image size, we conducted the following experiment. Following the methodology described in the "hyperparameter search and model training" section, we trained three CNN+RNN models using our dataset, with input images resized to $50 \times 50$, $100 \times 100$, and $200 \times 200$ pixels. We then simulated 200 students connected to the infrastructure with a single instructor. The assistant was run for 30 s under three

different configurations, each corresponding to one of the three image resolutions. In each case, the assistant used the corresponding model to perform predictions. We then measured how many images the assistant was able to process during the time window to assess whether real-time prediction was feasible for each configuration.

The instructor's computer was a 3.6 GHz Intel Core i7-7820X (8 cores/16 threads) with 16 GB of DDR4 3200 MHz RAM, running the latest version of Windows 11 Pro (24H2 update). Students used various desktop and laptop computers, all running Windows 11. The models were deployed on the Dell PowerEdge T630 server described earlier.

## Interpretability

To understand how the models make predictions, we visualize the discriminative features of the input sequences using gradient-weighted class activation mapping (Grad-CAM) (*Selvaraju et al., 2017*). Grad-CAM highlights the regions of input images that the model relies on for its predictions, providing insight into why certain activities are flagged as potentially fraudulent. It produces heatmaps that emphasize the most influential areas of an image with respect to a specific output class (*e.g.*, cheating). Grad-CAM is particularly suited for CNN-based models, as it visualizes the spatial regions within the feature maps that contribute most to the predicted class. In our experiments, we apply Grad-CAM to the final convolutional layer of the ANN architectures described in this section.

## Instructor satisfaction

After selecting the model with the highest $F_2$-score, we evaluated the entire assistant system (Fig. 3) in an exam involving 51 students, utilizing the CNN+RNN model (Fig. 4). The students were tasked with completing a programming activity within 50 min, using computers connected to the Internet. They were informed in advance about the types of activities that would be flagged as fraudulent.

Five instructors remotely monitored the students with the remote synchronous platform and our assistant, after receiving instructions on how to use both tools. Their task was to identify any fraudulent activities performed by the students during the exam. The students were not physically supervised in the laboratory room, as no instructors were present during the exam.

After the exam, the instructors were asked to complete the following questionnaire to evaluate their satisfaction with the assistant system. For each question, they were asked to indicate their level of agreement on a scale from 1 (completely disagree) to 5 (completely agree):

1. *The assistant was easy to use during the exam.*
2. *The assistant helped me to detect potentially fraudulent activities that I had not noticed by just looking at the monitorization system.*
3. *The system provided timely alerts that allowed me to react quickly to suspicious activities and check on that with the monitorization system.*
4. *I successfully filtered out false fraudulent activities by just using the assistant and the remote synchronous infrastructure.*

5. *After using both the monitoring system and the assistant, I think the assistant accurately identifies fraudulent activities.*

6. *The assistant's warnings about fraudulent activities were well-organized and easy to follow.*

7. *The assistant helped me prioritize which activities required my immediate attention.*

8. *The assistant significantly improved my ability to detect fraudulent activities during the exam.*

9. *Using the assistant made the monitoring process less stressful and more manageable.*

10. *Overall, I am satisfied with the performance of the assistant during this exam.*

The assistant logged all potentially fraudulent activities, including those deleted by the instructor, while the five instructors recorded all the fraudulent activities they identified. After the exam, we carefully reviewed all the recorded frames to determine the actual number of fraudulent activities performed by the students.

## RESULTS

Figure 5 shows the number of images relative to the number of users (200) yet to be processed by the assistant over 30 s for image sizes of $50 \times 50$, $100 \times 100$, and $200 \times 200$ pixels. We can see that $50 \times 50$ is the only image size that allows the assistant to perform predictions in real time. The other two configurations result in a monotonically increasing backlog of unprocessed images, indicating that the assistant cannot keep up with the data flow and therefore fails to operate in real time under those conditions. The $F_2$-score improvements for the $100 \times 100$ and $200 \times 200$ models were 0.52% and 0.61%, respectively, showing no significant advantage over the $50 \times 50$ model.

Table 4 summarizes the performance of the best-performing models. The ANN architecture depicted in Fig. 4—which integrates CNN, RNN, and dense layers—achieves the best results using the hyperparameters listed in Table 1. Statistical comparisons using paired $t$-tests ($\alpha = 0.05$) reveal no significant differences between this model and the EfficientNetB0 transfer learning approach in terms of accuracy, precision, $F_1$-score, or $F_2$-score (all $p \geq 0.05$). However, the hybrid CNN+RNN architecture achieves significantly higher recall ($p < 0.05$).

To further assess the performance difference between the CNN+RNN model and EfficientNetB0, we applied McNemar's test on the test set. The CNN+RNN model correctly classified 114 instances that EfficientNetB0 misclassified, while EfficientNetB0 correctly classified 96 instances that the CNN+RNN model misclassified. Based on these values, McNemar's test yielded a chi-square statistic of 1.376 and a $p$-value of 0.2407. Thus, we cannot reject the null hypothesis ($p \geq 0.05$), indicating that the difference in performance between the two models is not statistically significant. McNemar's test reveals statistically significant performance differences between the CNN+RNN model and the other evaluated models.

Despite their comparable classification performance, we selected the CNN+RNN model for integration into our assistant. In addition to its significantly higher recall, a key advantage lies in model complexity: the ANN with EfficientNetB0 requires 3.4 million

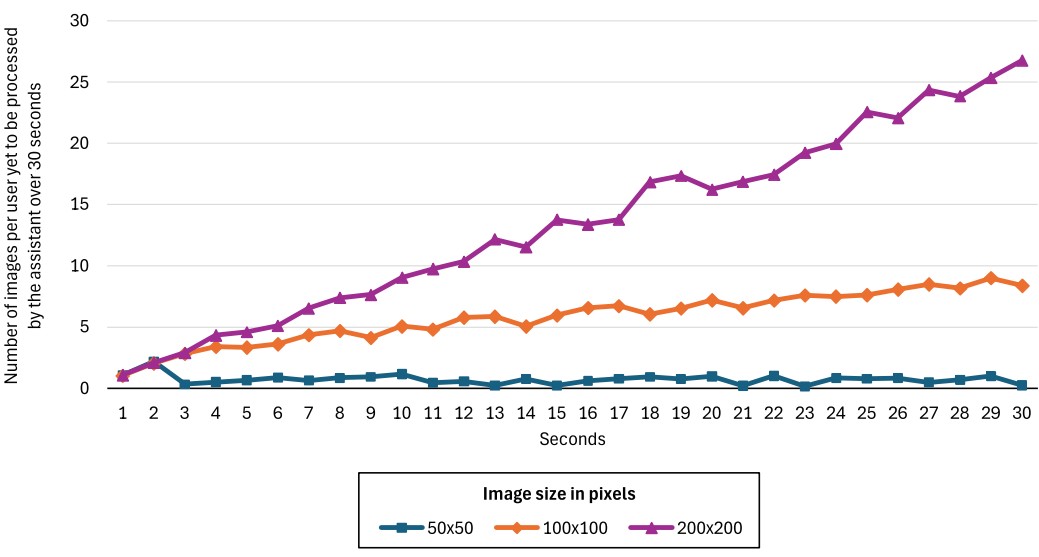

**Figure 5 Number of images per user yet to be processed by the assistant over 30 s for image sizes of 50 × 50, 100 × 100, and 200 × 200 pixels.**

**Table 4 Models with the highest performance metrics for the given methodology.**

| Model | Accuracy | Precision | Recall | $F_1$-Score | $F_2$-Score | Number of Parameters (millions) |
|---|---|---|---|---|---|---|
| CNN+RNN | **0.9518** | **0.9331** | **0.9710** | **0.9420** | **0.9592** | 1.3 |
| EfficientNetB0 | **0.9547** | **0.9352** | 0.9580 | **0.9465** | **0.9534** | 4.7 |
| MobileNetV2 | 0.9314 | 0.9010 | 0.9362 | 0.9183 | 0.9289 | 2.9 |
| NASNetMobile | 0.9007 | 0.8123 | 0.9482 | 0.8750 | 0.9175 | 4.8 |
| ResNet50 | 0.8847 | 0.8055 | 0.9147 | 0.8566 | 0.8906 | 24.6 |
| VGG16 | 0.8891 | 0.8259 | 0.9064 | 0.8643 | 0.8891 | 14.8 |
| VGG19 | 0.8686 | 0.8123 | 0.8718 | 0.8410 | 0.8592 | 20.1 |
| Optical flow (Farneback) | 0.6759 | 0.7052 | 0.4164 | 0.5023 | 0.4393 | 1.3 |

**Note:**
Bold values indicate the best performance for each metric. Significant differences are computed using a paired $t$-test ($\alpha = 0.05$); ties ($p \geq 0.05$) are shown with multiple bold values. All 95% confidence intervals are below 2%. The final column lists the number of trainable parameters (in millions).

more parameters—2.6 times more than the CNN+RNN model—making the latter a more efficient choice.

Figure 6 illustrates the impact of various factors on the $F_2$-score:

– The best performance was achieved with a configuration that included data augmentation, increased weighting of the positive class during training, and the use of the binary cross-entropy loss function.
– The evaluation without data augmentation highlights the critical role of the augmentation process described in the Methodology section, showing an average $F_2$-score improvement of 31.3%.
– Assigning double weight to the positive class (fraudulent activities) led to an average $F_2$-score increase of 4.35%.

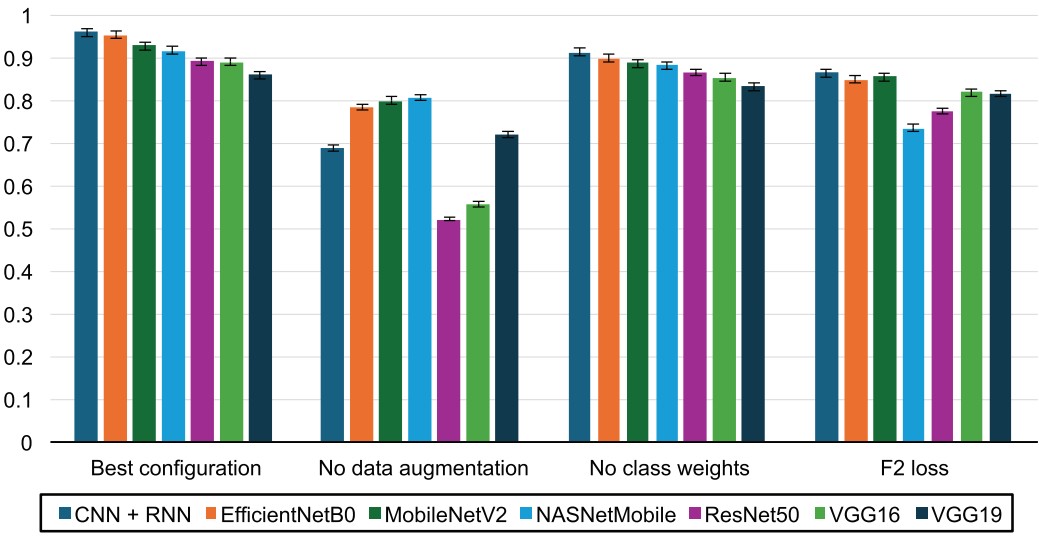

**Figure 6 Impact of transfer learning, data augmentation and class weights on prediction performance.**

– In contrast, the $F_2$ surrogate loss function proposed by *Lee, Yang & Yoo (2021)* resulted in a 12% decrease in performance compared to binary cross-entropy.

During the exam, four students out of 51 undertook fraudulent activities. They were rapidly admonished by a lecturer *via* a popup message sent through the remote synchronous infrastructure (Fig. 2). After that warning, no further fraudulent activity was observed from these students. The assistant successfully detected all four fraudulent actions, and the five lecturers handled the situations as per the established protocol. However, the assistant generated two false positives: one related to a Quick Watch popup window in Visual Studio and another to an XML file view in Chrome. In both cases, the instructors confirmed the absence of fraudulent activity by cross-referencing the assistant's alerts with the monitoring system. A post-exam manual analysis of the recorded videos revealed no additional fraudulent activity beyond those identified by the assistant.

Figure 7 presents example Grad-CAM visualizations from the CNN+RNN model when fraudulent activities are detected. In the first sequence, the student opens a web browser from the desktop. In the second example, the user switches from the Visual Studio IDE to a PDF reader while the Windows Start menu is open. The visualization shows that the CNN focuses on the PDF window, disregarding both the Start menu and the taskbar. In the third sequence, the student switches from Visual Studio to a side-by-side layout, where a web browser is visible. Again, Grad-CAM highlights that the prediction is driven by the part of the screen that does not display Visual Studio.

Table 5 presents the responses to the questionnaire designed to assess instructors' satisfaction with the assistant. For all but one question, the most frequent response (mode) was "completely agree". For five of the questions, all instructors selected "completely agree". The question with the lowest value ($Q_5$) had three responses of "agree" and two of "completely agree".

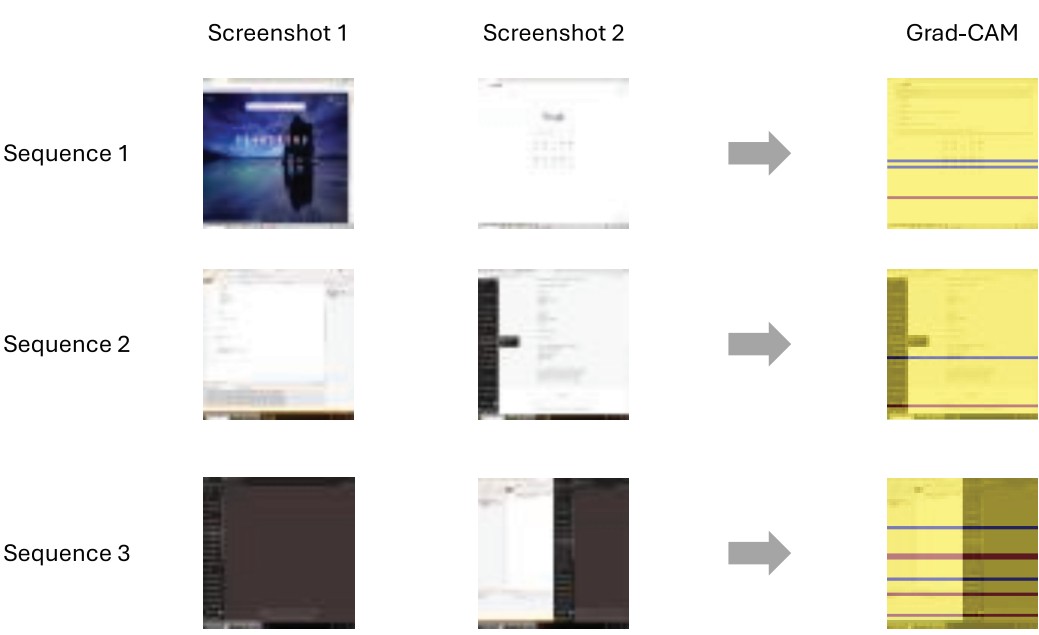

**Figure 7 Grad-CAM visualizations for three example sequences—the last two screenshots per sequence are visualized.**

**Table 5 Results of the questionnaire measuring the instructors' satisfaction.**

| Question | Mean | Standard deviation | Mode | Median |
|----------|------|--------------------|------|--------|
| $Q_1$ | 4.8 | 0.45 | 5 | 5 |
| $Q_2$ | 4.8 | 0.45 | 5 | 5 |
| $Q_3$ | 5.0 | 0.00 | 5 | 5 |
| $Q_4$ | 5.0 | 0.00 | 5 | 5 |
| $Q_5$ | 4.4 | 0.55 | 4 | 4 |
| $Q_6$ | 5.0 | 0.00 | 5 | 5 |
| $Q_7$ | 4.8 | 0.45 | 5 | 5 |
| $Q_8$ | 5.0 | 0.00 | 5 | 5 |
| $Q_9$ | 4.8 | 0.45 | 5 | 5 |
| $Q_{10}$ | 5.0 | 0.00 | 5 | 5 |

**Note:**
1 = completely disagree, 2 = disagree, 3 = neither agree nor disagree, 4 = agree, 5 = completely agree.

We conducted an analysis of our system's behavior under network stress using the configuration environment described in the "image size" subsection of the Methodology section. To simulate network latency, we employed the Linux Traffic Control subsystem (`tc`) to introduce delays of 50, 100, and 500 milliseconds in the connection between the instructor's computer and the web API hosting the predictive model. On average, it took 111.9, 225.2, and 1,030.1 milliseconds, respectively, for the instructor's computer to receive model predictions. During a 30-s observation window, the number of images pending processing did not increase, indicating that our system can detect potentially fraudulent activities in real time, even under network stress conditions.

We also evaluated the end-to-end processing time per student. Using $50 \times 50$-pixel images, the hardware configuration described in the Methodology section, and the best-performing CNN+RNN model, the assistant required an average of 1.2 milliseconds to process an image sequence and receive a prediction—of which 0.2 milliseconds corresponded to the model's inference time. Under these conditions, the system is capable of making real-time predictions for up to 833 students. Without GPU acceleration, the end-to-end processing time per student increases to 2.9 milliseconds, allowing the system to scale up to 344 students. The energy consumption for processing data from 200 students over the course of one hour was 107 watt-hours for the CPU-only configuration and 81.6 watt-hours for the GPU-enabled system. Although the GPU system has a higher peak power consumption, it was more energy-efficient overall due to spending more time in an idle state. At full utilization, the GPU configuration consumes 44.12% more power than the CPU-only setup.

## DISCUSSION

The exhaustive hyperparameter search enabled us to identify the optimal model for detecting potentially fraudulent activities within the given context. As shown in Table 4, transfer learning did not yield a significant performance improvement compared to our CNN+RNN neural architecture in Fig. 4. Additionally, Table 4 illustrates that the models with a larger number of parameters, except for EffcientNetB0, tend to perform worse. This may be attributed to the small image size ($50 \times 50$ pixels) used in our dataset, which likely causes the pre-trained models to overfit. The choice of small image dimensions was necessary to ensure the assistant could process all images generated by the remote synchronous infrastructure in real time—larger images could not be processed in real-time (Fig. 5). Furthermore, the CNN+RNN model, with the fewest parameters (1.3 million), facilitates faster inference, contributing to the system's ability to process frames in real time.

Our evaluation focused on the $F_2$-score metric, which places twice the weight on recall compared to precision. By prioritizing recall, the $F_2$-score emphasizes the detection of existing fraudulent activities. This approach ensured that our model detected all instances of fraudulent behavior during the exam, leaving no cheating unnoticed. While $F_2$-score also considers precision, the model managed to minimize false positives, with only two false positive detections observed in the evaluation. This outcome contributed to high instructor satisfaction, as reflected in the questionnaire responses. We believe this is the reason why the question with the lowest score (Question 5) addressed the model's prediction performance. Although the model achieved perfect recall, its precision was not flawless. Both the assistant and the monitoring system allowed instructors to quickly identify and address the false positives, as confirmed by the responses to Question 4.

Several strategies were employed to optimize the $F_2$-score performance of our model. Data augmentation played a significant role, leading to a 31.3% improvement in this metric. This technique not only enriched the training data but also helped balance the training set, yielding very favorable results. Additionally, increasing the weight of the positive class during training resulted in a further 4.35% gain in the $F_2$-score, enabling the

model to successfully detect all fraudulent activities in the exam. We also explored using a differentiable $F_2$ surrogate loss function (*Lee, Yang & Yoo, 2021*), but it did not outperform the binary cross-entropy loss.

An important feature of the assistant is its ability to facilitate the rapid detection of cheating activities. This allows instructors to promptly warn students about potential consequences, thereby preventing the continuation of fraudulent behavior. Such real-time intervention is undoubtedly more effective than merely storing frames and penalizing students after the exam has concluded.

The Grad-CAM visualizations help to interpret the model's decision-making process, highlighting regions associated with fraudulent activity—typically triggered by the appearance of a new window or popup message of any size. In the absence of such changes, the model does not flag any suspicious behavior. To further assess model interpretability, we conducted an error analysis based on the confusion matrix, which revealed 1.4% false negatives and 3.4% false positives. All false positives involved windows that students might legitimately open, such as text editors, XML and text file views in an Internet browser, or benign Visual Studio popups—none of which constituted cheating. The false negatives, though fewer, primarily involved cases where students viewed unauthorized PDF documents within the IDE or when small popup messages from instant messaging applications—not included in the training data—appeared. This analysis highlights the need to enrich the training dataset with more diverse examples of both legitimate and fraudulent behavior to improve the model's accuracy and reduce misclassifications.

Our system serves as an assistant for detecting potentially fraudulent activities in online exams, though human intervention remains necessary. It is not a standalone cheating detection system but rather a tool to assist instructors in monitoring student activity, especially in exams with large numbers of students. The optimization of the $F_2$-score ensures that the system is highly efficient in identifying potential cheating actions. However, the responsibility of verifying whether an activity is indeed fraudulent lies with the instructor, who must then take the appropriate actions, such as issuing a warning through the remote synchronous platform.

One advantage of our system is that it requires no additional hardware beyond a computer with Internet access. It focuses on analyzing students' activities on their own computers. However, it does not monitor the student's surrounding environment. Therefore, it can be effectively complemented by other online proctoring systems that use additional hardware to observe students and their environments during exams, but typically do not detect fraudulent use of unauthorized online resources.

As previously noted, our system is designed to assist instructors in detecting fraudulent activities, though it does not guarantee that such activities are entirely prevented. Various forms of online cheating have been documented, including student collusion and contract cheating (*Cleophas et al., 2021*). Our remote infrastructure provides alerts when multiple connections are detected for the same user. Additionally, it retrieves data from the VPN server and warns the instructor if the student's original IP address is located outside the university's geographic region (*Castro et al., 2023*). In either case, the instructor contacts

the student *via* the web conferencing platform to request an explanation. While these measures do not definitively prevent cheating, they serve as effective deterrents.

The current implementation of the system is tailored to the specific course for which the model was trained. In that course, all students are required to use Visual Studio on the Windows operating system, making the trained model applicable only to this particular IDE and platform. However, the architectures presented in Figs. 2 and 3 are generalizable and can be used to assist in detecting fraudulent activities in online exams across different courses. The component that must be adapted for each new course is the predictive model in Fig. 3. One approach is to create a dataset of images, both with and without fraudulent sequences, and train the model as described in this article. While this method would allow the system to be customized for each course, it requires retraining the model for every new subject, which can be time-consuming and resource-intensive. Alternatively, a more efficient strategy is to download our pre-trained model and retrain only the last hidden dense layer for the specific course (*i.e.*, transfer learning). This approach would reduce the need for full retraining, thereby saving both time and computational resources. Furthermore, cross-institutional collaboration could be highly beneficial for expanding the diversity of our dataset and improving the generalizability of the system. By incorporating data from a wider range of institutions, platforms, and student populations, the model could be better adapted to varying exam settings and more robust against context-specific biases.

## Threat to validity

One limitation of our evaluation is the relatively small size of the exam test cohort (51 students), with only four instances of actual fraudulent behavior observed during the exam. While the experiment provided valuable insights into the practical functioning of the assistant system in a real-world setting, the limited number of participants and cheating cases restricts the generalizability of our findings. In particular, the small sample size does not allow for robust claims about the scalability or statistical reliability of the system across broader or more diverse student populations. Future studies involving larger cohorts, and a wider range of academic contexts are necessary to validate the system's effectiveness at scale and to ensure consistent performance under varying conditions.

## CONCLUSIONS

We show how an artificial neural network, consisting of CNN layers followed by RNN and dense layers, can assist instructors in detecting potentially fraudulent activities in online exams where Internet access is restricted. The proposed system analyzes sequences of screenshot frames (captured at one frame per second) to identify cheating behaviors, achieving an accuracy of 95.18% and an $F_2$-score of 94.2%. We explored several techniques to enhance the model's performance, including transfer learning, data augmentation, class-weight adjustments during training, and an alternative loss function. Notably, data augmentation and increasing the weight of the positive class significantly improved the model's performance. The primary evaluation metric used was the $F_2$-score, which prioritizes the model's ability to correctly identify fraudulent activities (*i.e.*, recall) over its

precision. Our system provides valuable assistance in detecting potentially fraudulent activities in online exams, though human intervention remains necessary. In post-deployment feedback, instructors completely agree they were satisfied with the performance of the assistant.

As noted, the trained model is tailored for the specific course it was designed for. Future work will explore the extent to which this pre-trained model can be adapted to other courses by retraining only the final dense layer. Additionally, it would be valuable to investigate the development of a meta-model capable of adapting to various courses (*Rodriguez-Prieto, Pato & Ortin, 2025*). Meta-learning techniques, such as few-shot learning, could be explored to allow the system to generalize effectively across different subjects without the need for extensive retraining (*Álvarez-Fidalgo & Ortin, 2025*).

All the code used in our research, including the training, hyperparameter search, and fine-tuning of the models, the ready-to-use serialized CNN+RNN model, the implementation of the assistant, the image sequence labeler, and the Flask Web API, along with the training and test datasets and the data files generated during the experiments presented in this article, are freely available for download at https://reflection.uniovi.es/download/2025/assistant. This study was approved by the Institutional Review Board of the University of Oviedo (approval number: 25_RRI_2024). All participants provided informed consent prior to participation; the consent form is available for download at the same URL.

### Funding

This work has been funded by the Government of the Principality of Asturias, with support from the European Regional Development Fund (ERDF) Under Project IDE/2024/000751 (GRU-GIC-24-070). Additional funding was provided by the University of Oviedo through its Support for Official Research Groups (PAPI-24-GR-REFLECTION). The funders had no role in study design, data collection and analysis, decision to publish, or preparation of the manuscript.

### Grant Disclosures

The following grant information was disclosed by the authors:
Government of the Principality of Asturias.
European Regional Development Fund (ERDF) Under Project: IDE/2024/000751 (GRU-GIC-24-070).
University of Oviedo through its Support for Official Research Groups: PAPI-24-GR-REFLECTION.

### Competing Interests

The authors declare that they have no competing interests.

## Author Contributions

- Francisco Ortin conceived and designed the experiments, performed the experiments, analyzed the data, performed the computation work, prepared figures and/or tables, authored or reviewed drafts of the article, and approved the final draft.
- Alonso Gago performed the experiments, performed the computation work, authored or reviewed drafts of the article, and approved the final draft.
- Jose Quiroga performed the experiments, authored or reviewed drafts of the article, and approved the final draft.
- Miguel Garcia performed the experiments, authored or reviewed drafts of the article, and approved the final draft.

## Data Availability

All the code used in our research, including the training, hyperparameter search, and fine-tuning of the models, the ready-to-use serialized CNN + RNN model, the implementation of the assistant, the image sequence labeler, and the Flask Web API, along with the training and test datasets and the data files generated during the experiments are available in the Supplemental Files and at Zenodo: Ortin, F. (2025). Dataset used in the article "a machine learning assistant for detecting fraudulent activities in synchronous online programming exams" (1.0) [Data set]. Zenodo. https://doi.org/10.5281/zenodo.15881027.

## Supplemental Information

Supplemental information for this article can be found online at http://dx.doi.org/10.7717/peerj-cs.3159#supplemental-information.

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
