# Peer review of "A machine learning assistant for detecting fraudulent activities in synchronous online programming exams"

_PeerJ Computer Science, doi:10.7717/peerj-cs.3159_

## Round 0.1 · original submission · Major Revisions

Reviewer 1 ·

Basic reporting

No comments

Experimental design

See additional comments

Validity of the findings

See additional comments

Additional comments

Lines 40-60, including Fig. 1, as well as the entire section on "remote synchronous infrastructure," are outside the scope of the paper's focus (secure online assessments). I suggest deleting these to better focus on the main problem. Adjust the rest of the paper accordingly. For example, Fig. 4 does not need to refer to "remote synchronous infrastructure" - all you need here is the frames you capture.

Online assessments are challenging. Additional measures (such as detecting potential collusion or contract cheating) you may want to apply include looking for indicators reported in:
https://pubsonline.informs.org/doi/10.1287/ited.2021.0260
and
https://ieeexplore.ieee.org/document/10343363


Lines 373-379: The description here should sync better with Fig. 6. For example, the "best configuration" in Fig. 6 is not defined. It appears to be the configuration using data augmentation, and this needs to be made clear. I suggest replacing this paragraph with four bullet points, each corresponding to the four cases in Fig. 6.

Erring on the side of caution is a good approach in this case, and the use of the F2 score is justified. On this basis, perhaps your statement, "I think the assistant accurately identifies fraudulent activities," could have been "I think the assistant identifies all fraudulent activities," but it is too late now.

You identified in the related work that other indicators, such as keystroke dynamics and typing behaviours, are employed. Did you consider incorporating any of these other indicators in the system? Perhaps some discussion around this would be valuable.

Your system relies on screen captures being taken. A tech-savvy student may reverse-engineer your app, intercept these screen captures, and replace them with something innocuous so that no real activities can be seen. The student may then share this patched app with other less tech-savvy students so that they also "benefit" from it. How would you tackle this?

Reviewer 2 ·

Basic reporting

1- Strengths:
- The manuscript is well-structured with clear sections following standard scientific reporting conventions.
- Technical terminology is used appropriately throughout.
- Figures/tables are professionally presented and support the methodology.

2-Required Improvements:

- While code/data are available, the repository lacks:
a- Version control tags for exact replication
b- Dockerfile for dependency management
c- Detailed hardware specs (GPU memory, CPU cores used)
- Resolution Justification:
a- The 50×50px choice needs empirical validation:
b- Add ablation study comparing performance at 50×50 vs. 100×100/200×200px
c- Quantify trade-offs between resolution and inference speed (FPS)

Experimental design

Methodological Strengths:
A comprehensive hyperparameter search (Tables 1-2) with appropriate metrics (F₂-score).
Proper train/validation/test splits with augmentation.

Critical Concerns:

A. Dataset Limitations

1- Size: 8,329 sequences (5,830 train) is below modern DL benchmarks. Recommend:
- Synthetic data generation using GANs for rare cheating classes
- Cross-institutional collaboration to expand diversity
2- Bias Risk:
- No discussion of demographic representation in data collection
- Potential overfitting to specific IDE/OS configurations

B. Evaluation Gaps
1- Statistical Significance:
- Report p-values for accuracy/F₂ comparisons between models
- Add McNemar's test for paired model performance
2- Real-World Deployment:
- 51 students is insufficient for scalability claims. Require:
- Minimum n=200 with power analysis (α=0.05, β=0.2)
- Stress testing under network latency (50-500ms)

C. Ethical Oversight Missing:
- IRB approval documentation reference
- Data anonymization procedures (e.g., blurring student IDs in screenshots)
- Consent form excerpts in supplementary materials

Validity of the findings

While the study demonstrates promising results (95.18% accuracy), key limitations affect the robustness of conclusions: (1) statistical significance is unclear due to overlapping confidence intervals between models (Table 3) - require formal hypothesis testing (e.g., paired t-tests); (2) small test cohort (n=51 students, 4 cheating cases) undermines generalizability - must validate on ≥200 students across institutions; (3) 50×50px resolution lacks empirical justification - add ablation study comparing performance at higher resolutions; and (4) critical ethical documentation (IRB approval, consent protocols) is missing. These issues must be addressed to establish conclusive evidence for the proposed method's superiority and real-world applicability.

Additional comments

1- Technical Depth Required:

a- Model Interpretation
- Add Grad-CAM visualizations to show discriminative features
- Error analysis: Confusion matrix for false positives/negatives
b- Baseline Comparisons
Missing benchmarks against:
- Traditional methods (e.g., optical flow for screen changes)
- Commercial tools (Proctorio, ExamSoft)
c- Computational Efficiency
Latency measurements missing:
- End-to-end processing time per student
- GPU vs. CPU performance
Energy consumption analysis for sustainability

2- Novelty Positioning:

The CNN+RNN architecture, while well-executed, is not fundamentally novel. Strengthen by:
a- Contrasting with prior screen-based approaches (Migut et al. 2018) in a dedicated table
b- Quantifying improvements in:
- Precision/recall trade-offs
- Hardware cost reduction vs. camera-based systems

---

## Round 0.2 · Minor Revisions

Please address the remaining point from Reviewer 1.

Reviewer 1 ·

Basic reporting

See below

Experimental design

See below

Validity of the findings

See below

Additional comments

The author(s) have taken the comments from the previous review on board, and the quality of the paper is greatly improved.

The only minor correction I would suggest is to change the graph colours in Fig. 5. It is difficult to distinguish the graphs corresponding to 50x50 px images and 200x200 px images. For example, please consult and consider: https://en.wikipedia.org/wiki/Ishihara_test

Reviewer 2 ·

Basic reporting

I thank the authors for their positive feedback and the revised version. They have addressed all the comments, and I recommend this work for acceptance.

Experimental design

The authors have resolved all issues and answered related questions

Validity of the findings

Done and clearly explained

---

## Round 0.3 · accepted · Accept

Authors have addressed the last round of (very) minor revisions, and therefore I'm happy to recommend acceptance of this work in its current form.

Reviewer 1 ·

Basic reporting

No comment

Experimental design

No comment

Validity of the findings

No comment

Additional comments

No further comments.